# Occupancy Grid-Based AUV SLAM Method with Forward-Looking Sonar

**Xiaokai Mu** [1,2,*]**, Guan Yue** [1]**, Nan Zhou** [2] **and Congcong Chen** [1]

[1] Science and Technology on Underwater Vehicle Technology Laboratory, Harbin Engineering University, Harbin 150001, China; yue_guan@hrbeu.edu.cn (G.Y.); chencongcong@hrbeu.edu.cn (C.C.)

[2] Qingdao Innovation and Development Center, Harbin Engineering University, Qingdao 266000, China; 1998_zn@hrbeu.edu.cn

[*] Correspondence: muxiaokai@hrbeu.edu.cn

**Abstract:** Simultaneous localization and mapping (SLAM) is an active localization method for Autonomous Underwater Vehicle (AUV), and it can mainly be used in unknown and complex areas such as coastal water, harbors, and wharfs. This paper presents a practical occupancy grid-based method based on forward-looking sonar for AUV. The algorithm uses an extended Kalman filter (EKF) to estimate the AUV motion states. First, the SLAM method fuses the data coming from the navigation sensors to predict the motion states. Subsequently, a novel particle swarm optimization genetic algorithm (PSO-GA) scan matching method is employed for matching the sonar scan data and grid map, and the matching pose would be used to correct the prediction states. Lastly, the estimated motion states and sonar scan data would be used to update the grid map. The experimental results based on the field data have validated that the proposed SLAM algorithm is adaptable to underwater conditions, and accurate enough to use for ocean engineering practical applications.

**Keywords:** AUV; SLAM; grid map; scan matching

## 1. Introduction

Autonomous Underwater Vehicle (AUV) is an essential piece of equipment for ocean exploration, and accurate navigation is crucial for AUV to perform various missions. Achieving precise localization in an unknown and complex environment is challenging for the development of AUV. Simultaneous localization and mapping (SLAM) is an active method for this scenario where it can realize localization while constructing the environment map. Therefore, the research on the SLAM technology of AUV is of great significance for ocean engineering applications.

SLAM needs sensors that can explore the external environment. The sensors used in underwater SLAM mainly include the camera, laser scanner, side-scan sonar, multi-beam sounder, and forward-looking sonar [1,2]. An amount of research has been focused on the above SLAM technology. The underwater camera can obtain a high-definition image, and the accuracy of AUV motion can be estimated through image matching. However, the optical signal is sensitive to the water quality, which leads to the detection range being not long, especially when the water is turbid [3,4]. The laser scanner can obtain high-resolution point cloud data. The point cloud data can accurately describe the contour information of the environment. A large amount of data leads to a high computational complexity when applied to a large-scale map. Therefore, the SLAM using a laser scanner is suitable for small scenes [5,6]. The side-scan sonar transmits sound waves to the seafloor and receives the echo. According to the wave intensity, the side-scan sonar can obtain seafloor geomorphic features. Because of the principle of side-scan sonar, it needs to move forward continuously to obtain the geomorphic image. Therefore, SLAM can only be realized by planning the path that can repeatedly observe the target [7,8]. The working model of the multi-beam sounder is similar to that of the side-scan sonar. It can obtain the terrain information when the AUV is moving forward. Similarly, SLAM can only be achieved by planning the path

of the repeated course [9,10]. Forward-looking sonar transmits sound waves and estimates the front environment information according to the echo intensity. The forward-looking sonar can get a relatively low resolution but continuous target information during the cursing state of the AUV, so it is suitable for the SLAM navigation system in a complex environment [11–13].

Map construction is one of the core functions of SLAM. To realize mapping based on the environment sensor, a proper description of the environment is needed. The main presentation includes a feature map, a topology map, and a grid map [14,15]. Among them, The description of the feature map is the most intuitive, and it can better describe the appearance information of the observed features [16,17]. However, both point features and line features are more dependent on the sensor's accuracy, and the measurement noise of the sensor itself will also seriously interfere with the feature extraction effect. The designed SLAM navigation system obtains environmental information through the forward-looking sonar. Due to the low signal-to-noise ratio of the sonar image, the error of the feedback distance and angle information is relatively large, and the SLAM based on the feature map performs poorly in a complex underwater environment. A topology map is another way to abstract the representation of the environment. The method uses nodes to represent environmental information collected by sensors and record the relationships between different nodes, including the relative position, orientation, overlap, and inclusion [18,19]. How to convert environmental information into nodes is the main problem in the topology map, especially in the complex underwater scenario.

By dividing the territory into small grids, the grid map can construct a rough map [20]. As each grid represents whether there is a target in this region, this method can work appropriately with a relatively low accuracy sensor. Especially for point features, raster images occupy fewer system resources. Since the grid composition method divides the environmental map into small grids of each region, the requirements for the accuracy of environmental sensors are relatively low, so it is suitable for sensing using sensors such as forward-looking sonar. Compared to the feature map, this method could perform well with insufficient storage space and is convenient for future AUV path planning [21]. Therefore, it is suitable for underwater map construction.

The main grid map-based SLAM methods include Gmapping, Hector SLAM, Cartographer, etc. [22,23]. Despite many successful applications of the above SLAM methods, they are still not suitable for AUV navigation. Gmapping is based on a particle filter, which needs more computing resources in large scenarios [24–26]. Hector SLAM only employs sensing data and does not rely on other sensors. Therefore, it is unsuitable for the AUV SLAM system [27,28]. Like the Hector SLAM, the Cartographer uses sensing data to correct the AUV motion. Accuracy sensing is required, but the other sensors of the AUV navigation system cannot be fully utilized [29–31].

In this paper, a computationally inexpensive and practical AUV SLAM navigation method is studied. The occupied grid map is employed to describe the environment based on the data from the forward-looking sonar. The algorithm employs a Kalman filter for state estimation. The sonar scan data is used to build a grid scan map and extracts a local submap from the environment map according to the area of the grid map. The grid scan map and the local submap are matched by the particle swarm optimization genetic algorithm (PSO-GA). The matched pose is selected as the observation of the filter to correct the navigation information. The experimental results based on AUV actual field data verified the effectiveness of the proposed SLAM method. The remainder of this paper is organized as follows: Section 2 introduces the fundamental of the studied SLAM system, including the AUV platform and the occupancy grid map. The details of the proposed algorithm are presented in Section 3. The experimental results based on AUV field data are analyzed in Section 4. Finally, Section 5 summarizes the principal conclusion of this work.

## 2. Fundamental of the Studied SLAM System

### 2.1. Platform Introduction

The structure of the experiment platform is shown in Figure 1. The sensors and devices of the navigation system include a global position system (GPS), Doppler velocity log (DVL), inertial measurement unit (IMU), and depth meter (DM). Figure 2 is the deployment scenario of the experiment platform. Due to the advantages of forward-looking sonar in a complex environment, the SLAM system uses a head-mounted forward-looking sonar for environment sensing.

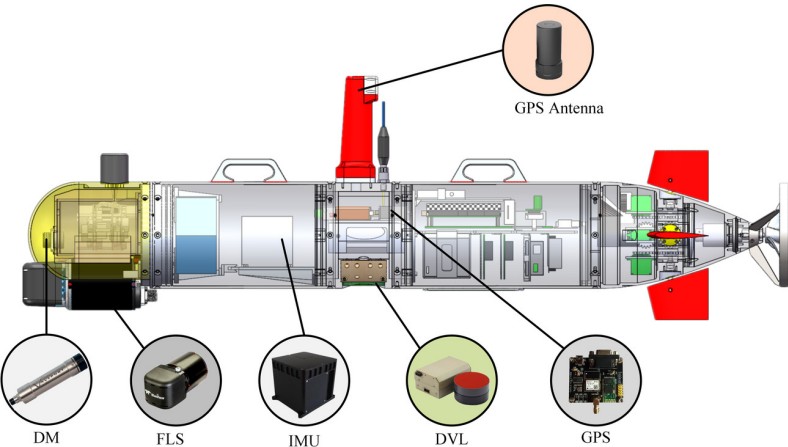

**Figure 1.** The structure of the experiment platform.

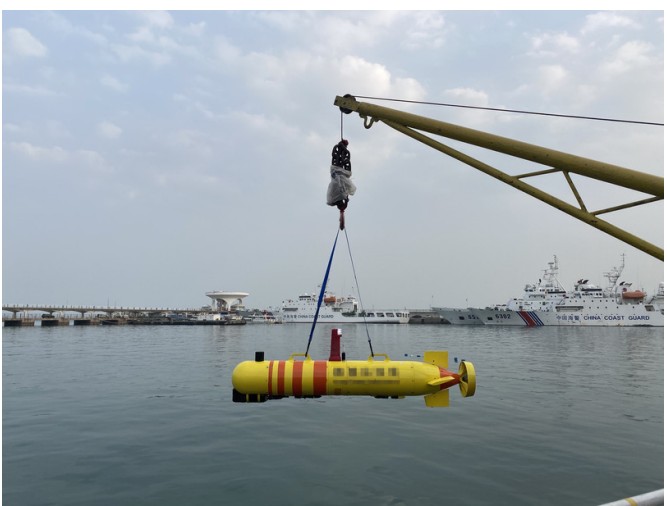

**Figure 2.** The deployment scenario of the experiment platform.

The GPS equipment includes the antenna and receiver. The antenna is on the top of AUV and can receive the satellite signals when the AUV floats on the surface. Then, the GPS receiver would calculate the position information according to the satellite signals. As the GPS position has no accumulation errors, the GPS information usually can be trusted when the AUV is on the surface.

The DVL is mounted at the bottom of the AUV. It can transmit short sound information, and the DVL can obtain the vehicle velocity information according to the Doppler shift. The velocity of DVL is relatively accurate, but it is easy to be interfered with by the surrounding environment.

The IMU is mounted in the middle of the vehicle. It contains three gyroscopes and accelerometer. According to the raw angular speed and accelerometer, the IMU can integrate precise attitude angle and yaw information of AUV. However, the accuracy of IMU will diverge with time.

The DM is a water pressure sensor. According to properties such as density and pressure value of seawater, it can obtain an accurate absolute depth. Consequently, the SLAM can be converted into a two-dimensional (2D) situation, and the following content is the 2D position estimation and map construction.

### 2.2. Occupancy Grid Map

As the underwater SLAM can be regarded as the position and mapping in the 2D horizontal plane, the map used by the SLAM system studied in this paper is a 2D occupancy grid map, which is divided into a finite number of units according to the scale. The occupancy grid map is expressed in Equation (1).

$$M = \{m_i\} \tag{1}$$

where $m_i$ represents the grid map with index $i$. The probability of occupation is expressed as $P(m_i)$. Since the occupation only includes whether grid cell is occupied or not, the probability of occupation is as follows.

$$p(m_i = 0) + p(m_i = 1) = 1 \tag{2}$$

The posterior probability represents the occupancy probability in the SLAM algorithm, as shown in Equation (3).

$$p(m_i | Z_{i:t}, X_{i:t}) \tag{3}$$

where $Z$ represents the observation and $X$ represents the vehicle state. The map is constructed as the occupancy probability of each grid cell, which is represented by binary Bayes. This method is expressed as the odds ratio of the occurrence and non-occurrence probability. As Equation (4) shows:

$$\frac{p(m_i = 1)}{p(m_i = 0)} = \frac{p(m_i = 1)}{1 - p(m_i = 1)} \tag{4}$$

During the process of SLAM, the posterior probability of the occupancy grid is constantly updated by the measurement, and the value of probability is always between [0, 1]. Therefore, the logarithm is introduced in the process of probability update, and the expression of the logarithm probability is as follows:

$$l_{t,i} = \log \frac{p(m_i | Z_{i:t}, X_{i:t})}{1 - p(m_i | Z_{i:t}, X_{i:t})} \tag{5}$$

The logarithmic probability of each grid cell is shown in Equation (6).

$$l_{t,i} = l_{t-1,i} + inverse\_sensor\_model(m_i, X_t, Z_t) - l_0 \tag{6}$$

where $l_{t,i}$ represents the logarithmic probability of grid cell $i$ at time $t$, and $l_0$ is the initial value of the logarithmic probability. The *inverse_sensor_model* also uses logarithmic form and is expressed as follows:

$$inverse\_sensor\_model\ (m_i, X_t, Z_t) = \log \frac{p(m_i | X_t, Z_t)}{1 - p(m_i | X_t, Z_t)} \tag{7}$$

The Bresenham algorithm is used to judge whether the grid cell is occupied. Figure 3 shows an example of the inverse measurement model. The gray grid cell indicates that this cell is not detected, the black grid suggests that this cell is detected, and the white grid indicates no target area.

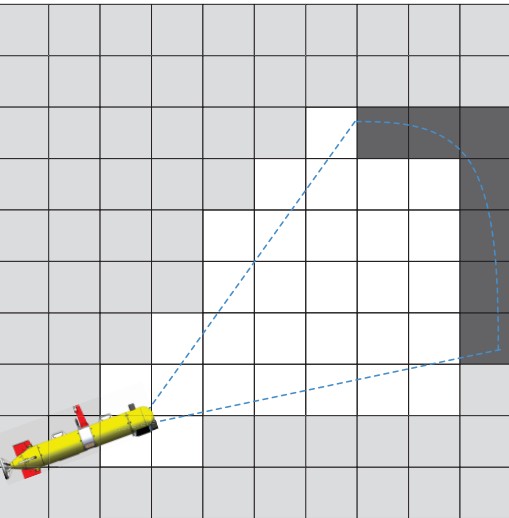

**Figure 3.** Example of the inverse measurement model.

The logarithmic probability of each cell can be updated according to Equations (6) and (7), and then converted into the occupancy probability. The conversion function is shown in Equation (8).

$$p(m_i \mid Z_{i:t}, X_{i:t}) = 1 - \frac{1}{1 + \exp(l_{t,i})} \tag{8}$$

The occupancy grid map represents the probability of the existence of targets in the cell. In the SLAM process, the probability needs to be binarized, and the cells with high possibility are considered the occupied cell, while the remaining cells are not occupied.

## 3. Occupancy Grid-Based AUV SLAM Method

This paper proposed an occupancy grid-based SLAM algorithm for AUV. Figure 4 shows the flow chart of the proposed method. This method employs an extended Kalman filter (EKF) to estimate the AUV state. According to the occupancy grid map method described in the previous section, the system first initializes the pose and map. As the data frequency of multi-beam forward-looking sonar is lower than that of other navigation sensors, the state and covariance of the system are predicted by the time update process of EKF. When the sonar data are updated, the feature points in the sonar data are extracted to map the detected feature into a scanning grid map, and the sub-map is extracted in the detected target area from the whole grid map. The scanning grid map and sub-map realize the pose estimation of AUV by the matching method, and the estimated AUV pose is used as the measurement to update the system state and covariance. Finally, the system transforms the sonar data into the navigation coordinate system according to the estimated pose.

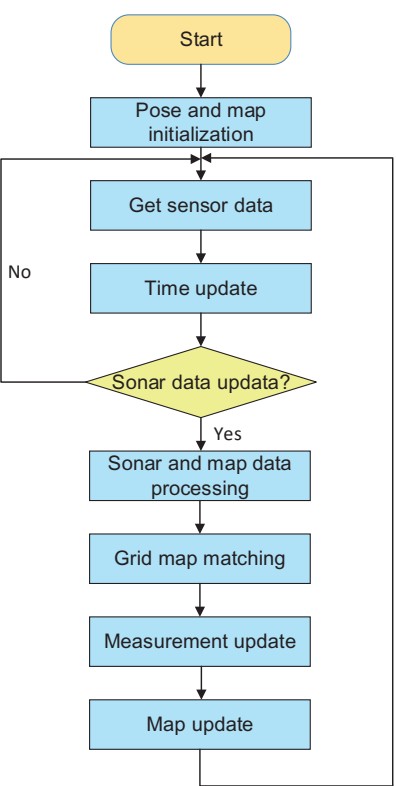

**Figure 4.** Occupancy grid-based AUV SLAM method.

### 3.1. Time Update Process

The dead reckoning algorithm is used to predict the state of the AUV system status. The orientation of IMU and velocity of DVL in the horizontal plane are employed for state estimation. In this study, the system state can be represented by the horizontal position and orientation. Equation (9) is the kinematics equation used for the time update.

$$X_{k+1} = f(X_k, U_k)$$
$$\begin{bmatrix} x \\ y \\ \psi \end{bmatrix}_{k+1} = \begin{bmatrix} x + u\cos(\psi_I)t - v\sin(\psi_I)t \\ y + u\sin(\psi_I)t + v\cos(\psi_I)t \\ \psi_I \end{bmatrix}_k \tag{9}$$

The system state of AUV is $X = [x, y, \psi]^T$ in the "north-east" navigation coordinate system, where $x$ and $y$ represent the vehicle location on the north and east axes, respectively, and $\psi$ is the orientation angle of the AUV, zero degrees to the north and positive clockwise. The control input is $U = [u, v, \psi_I]^T$, where $u$ and $v$ respectively represent the forward and lateral velocity of the DVL in the airborne coordinate system. $\psi_I$ is the orientation angle from IMU. $t$ is the time interval of the time update. Since the time update includes the system state and control input, the Jacobian matrix should be calculated. Equations (10) and (11) are the Jacobian matrix of the system state and control input, respectively.

$$F_X = \begin{bmatrix} 1 & 0 & 0 \\ 0 & 1 & 0 \\ 0 & 0 & 0 \end{bmatrix} \tag{10}$$

$$F_U = \begin{bmatrix} \cos(\psi_I)t & -\sin(\psi_I)t & -u\sin(\psi_I)t - v\cos(\psi_I)t \\ \sin(\psi_I)t & \cos(\psi_I)t & u\cos(\psi_I)t - v\sin(\psi_I)t \\ 0 & 0 & 1 \end{bmatrix} \tag{11}$$

The system state covariance can be calculated by Equation (12).

$$P_{k+1|k} = F_X P_k F_X{}^T + F_U Q F_U{}^T \qquad (12)$$

### 3.2. Sonar and Map Data Processing

The data of the sonar and map need to be processed before being used to optimize the AUV state. The processing process is as follows. As sonar data are noisy, effective features should be extracted. The forward-looking sonar can obtain the echo intensity from different beams. The intensity of the echo reflects the possibility of the target within the detection range. As the closer-range data are more reliable than the further data, the first point of each beam greater than the threshold is the feature point. The features extraction effect of sonar data is shown in Figure 5.

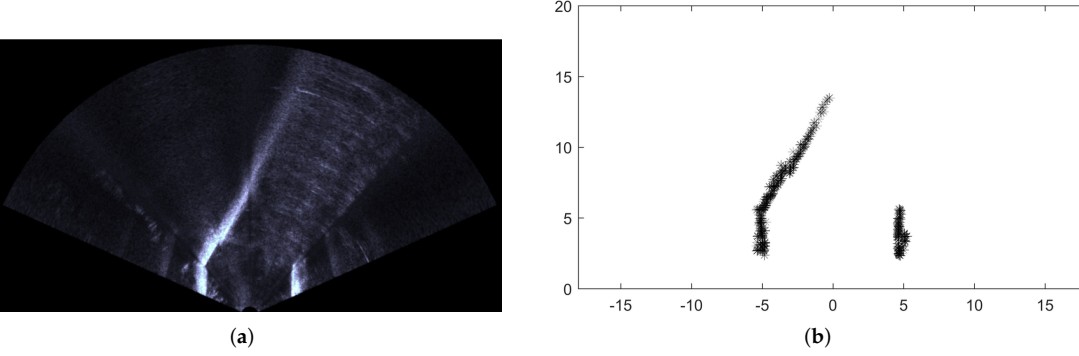

**Figure 5.** Features extraction of sonar data. (**a**) Raw sonar data. (**b**) Sonar data after feature extraction.

The sonar data are based on the airborne coordinate system, which needs to be converted to the navigation coordinate system. The rotation and translation formula is depicted in Equation (13).

$$\begin{bmatrix} T_{wx} \\ T_{wy} \end{bmatrix} = \begin{bmatrix} \cos(\psi) & -\sin(\psi) \\ \sin(\psi) & \cos(\psi) \end{bmatrix} \begin{bmatrix} T_{sx} \\ T_{sy} \end{bmatrix} + \begin{bmatrix} x \\ y \end{bmatrix} \qquad (13)$$

where $T_{wx}$ and $T_{wy}$ are the positions of feature points in the navigation coordinate system, and $T_{sx}$ and $T_{sy}$ are the positions in the sonar airborne coordinate system. The sonar scanning grid map is constructed according to the specific position of sonar feature points. The scope of each scan grid map constructed is much smaller than the overall map constructed by the SLAM system. The matching between the scanning map and the entire map would cost much unnecessary calculation. Therefore, an appropriate sub-map from the whole map should be extracted to improve the efficiency of the SLAM execution. The sub-map is selected by the distribution area of feature points from the scan data. Local submap by probability values are binarized to indicate whether grid cells are occupied or not.

### 3.3. Grid Map Matching Method

Due to the error in the AUV state estimation during the time update process, the scanning grid map is deviated from the local submap. The purpose of map matching is to find an appropriate rotation and translation from the scanning grid map to the local map. The exhaustion Method can solve this problem. However, to obtain an exact optimal solution, a large number of solutions needs to be set up in advance. Considering the calculation time and accuracy, it is difficult to obtain an accurate pose by the exhaustive method. At present, the branch and bound method is a general grid map matching algorithm, which has been successfully applied to land robots. Nevertheless, the measurement accuracy of sonar is much lower than that of sensors on land, and the branch-and-bound method can easily fall into optimal local values and results in the SLAM failure. Therefore, a global optimization strategy should be used for grid map matching of sonar data. The genetic algorithm is one of the global optimization methods [32,33]. According to the competition

mechanism between target groups, the individuals far away from the optimal solution would be eliminated, and crossover and mutation operations are carried out among the remaining populations. After the above iterative process, the most suitable individuals are selected as the optimal solution. Although the genetic algorithm can quickly converge the pose in the initial process, it is difficult to approach the optimal pose of AUV only through its mutation mechanism. To solve this question, particle swarm optimization (PSO) [34–36] was introduced into the mutation process of the genetic algorithm (GA). The flow chart of the developed PSO-GA is shown in Figure 6.

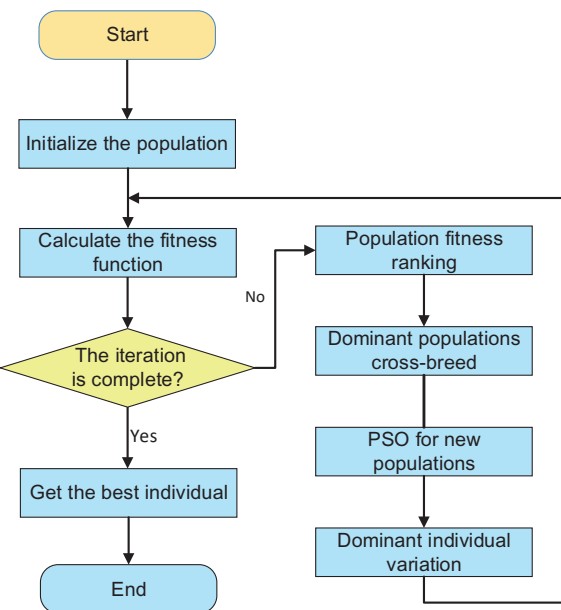

**Figure 6.** The grid map matching algorithm based on PSO-GA.

The population of the GA is initialized first. The gene of each individual includes the horizontal position and orientation $X_i = [x_i, y_i, \psi_i]^T$, where $i$ represents the index of the individual. The fitness function of the GA indicates the difference between the the scanning grid map generated by the individual and the local submap. The smaller the difference, the higher the fitness of the individual.

Half of the individuals with low fitness are abandoned during the iterative process, and the other would be selected for gene crossover. Each individual contains only three genes, and a new generation of the population is generated using a single point crossover operator.

Based on the new generation of the population, due to the randomness of the PSO, the fitness of some individuals will be improved, and these individuals will be selected as the variable individuals, while the rest of the individuals remain in the state before being optimized by the PSO. The new generation population optimized by the PSO continues to iterate until the optimal solution is found. Figure 7 shows the effect of grid map matching using the proposed PSO-GA algorithm.

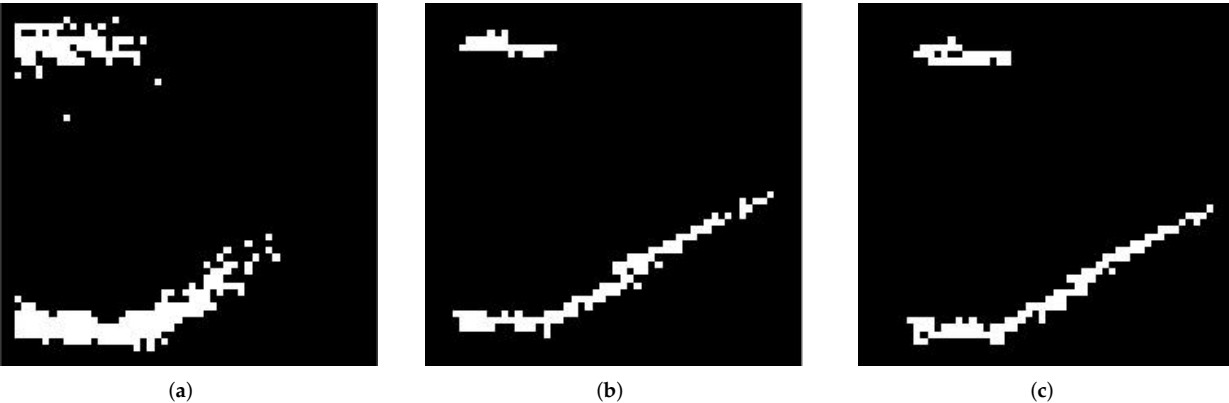

(**a**)                      (**b**)                      (**c**)

**Figure 7.** Effect of grid map matching. (**a**) Grid sub-map. (**b**) Scanning grid map. (**c**) Scanning grid map after matching.

### 3.4. Measurement Update Process

The map matching between the grid map and the local sub-map can find an appropriate AUV state, and the state is the observation in the measurement update process of the EKF. The expression of the observation is as follows:

$$Z = \begin{bmatrix} x & y & \psi \end{bmatrix}^T \tag{14}$$

Since the observation vector has a linear relationship with the system state, the observation model is linear, which is as follows:

$$H = I_{3\times3} \tag{15}$$

After the measurement update process, the EKF estimation would be used to transform the feature points to the navigation coordinate system and update the probability value of the grid map according to Equations (6) and (8). The updated grid map is used for the next map matching.

### 4. Experimental Result and Analysis

To verify the validity of the occupancy grid-based SLAM algorithm, experiments based on AUV field data were carried out. The field data were collected near Tuandao Bay wharf in Qingdao. Figure 8 shows the sea trail environment, including walls, bridge holes, rocks, and barges. The AUV moved clockwise on the water surface in the harbor, and the GPS position was used as the ground truth to evaluate the performance of different methods. The distribution of feature points collected by the forward-looking sonar is shown in Figure 9. The red lines in the figure are GPS trajectories, and the blue points are the feature points.

The experimental analysis included the performance of the proposed SLAM framework in different grid map matching methods, and under EKF and UKF.

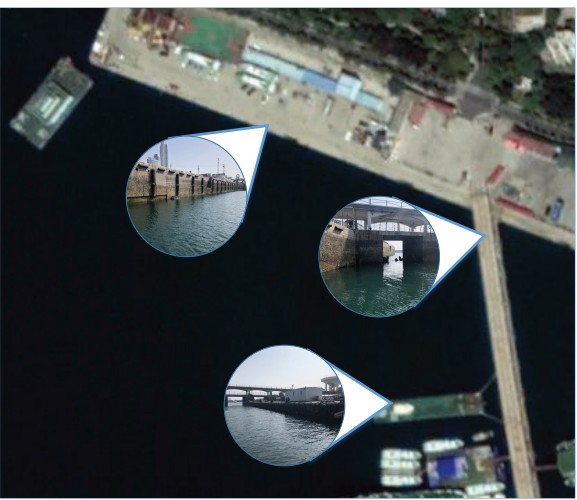

**Figure 8.** Sea trial environment.

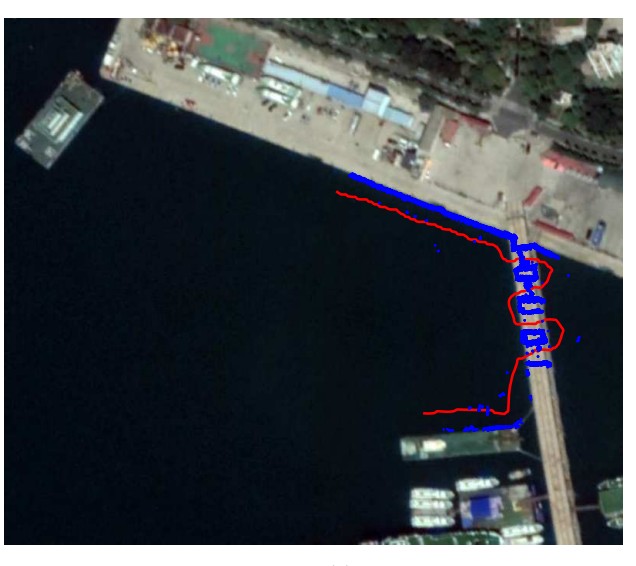

(**a**)

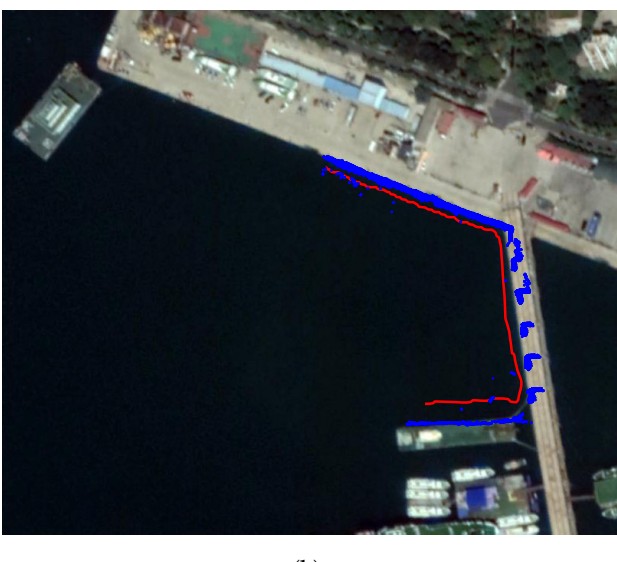

(**b**)

**Figure 9.** AUV real trajectory and sonar feature points. (**a**) AUV real trajectories and sonar feature points in Test 1. (**b**) AUV real trajectories and sonar feature points in Test 2.

*4.1. Experimental Results and Analysis under Different Grid Map Matching Methods*

In this section, the exhaustive method, GA, and the developed PSO-GA for grid map matching are examined. Figure 10 shows the AUV trajectories optimized by different matching methods. The black lines are the ground truth, red lines are the dead reckoning trajectory, blue lines are the SLAM trajectory based on the exhaustive method, green lines are the SLAM trajectory based on the developed GA, and purple lines are the SLAM trajectory based on the developed PSO-GA .

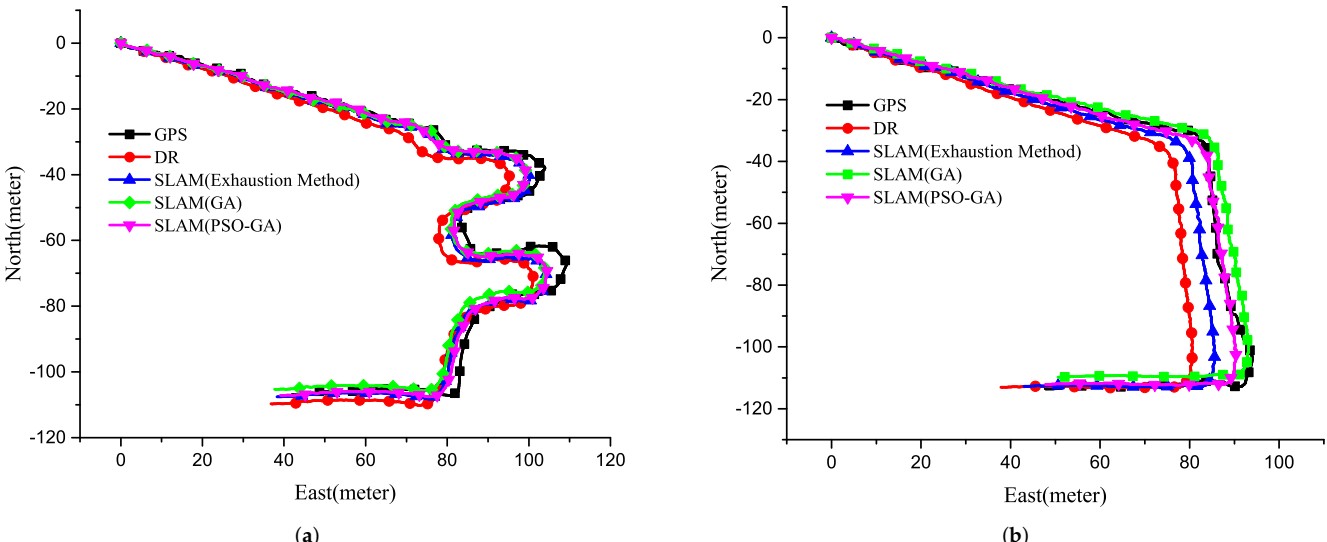

**Figure 10.** AUV positioning with different navigation methods. (**a**) AUV positioning with different navigation methods in Test 1. (**b**) AUV positioning with different navigation methods in Test 2.

In Figure 10, the position deviation of DR would increase faster without the SLAM algorithm. The abscissa represents the number of software runs, with a step size of 100 ms. Since the exhaustive method can approximately modify the AUV state, the position error accumulation can be limited to a certain extent. The exhaustive method cannot achieve the optimal matching, so there still exists a large error in the trajectory. The developed PSO-GA can ensure that the matching effect approximates the optimal solution, so as to obtain a better AUV trajectory.

Figure 11 shows the positioning errors by different algorithms, where the red lines are the error of dead reckoning, blue lines are the SLAM localization error based on the exhaustive method, green lines are the position error of SLAM based on the GA, and purple lines are the position error of SLAM based on the PSO-GA. The error results show that the developed PSO-GA can effectively reduce the positioning error. Table 1 shows the root mean squared error (RMSE) of the position under different algorithms.

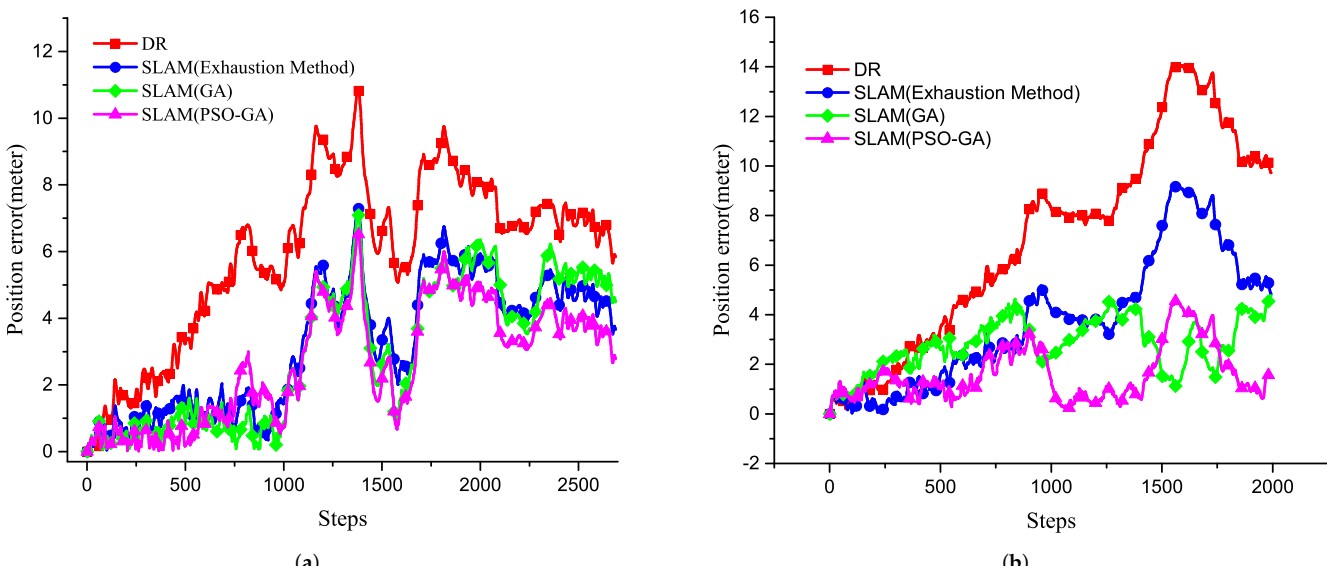

**Figure 11.** Localization error of different methods. (**a**) Localization error of different methods in Test 1. (**b**) Localization error of different methods in Test 2.

**Table 1.** RMSE of the position under different algorithms; unit: m.

|        | Dead Reckoning | SLAM (Exhaustive Method) | SLAM (GA) | SLAM (PSO-GA) |
|--------|----------------|--------------------------|-----------|---------------|
| Test 1 | 5.9904         | 3.2607                   | 3.0120    | 2.6900        |
| Test 2 | 7.1185         | 3.7900                   | 2.8062    | 1.6108        |

Figure 12 shows the probability grid map constructed by the SLAM algorithm based on PSO-GA. The probability of each grid is expressed as the gray value. By comparing the grid map and the feature points, it can be seen that the SLAM algorithm could effectively construct the map of the wall and bridge holes that the AUV passed. The underwater rocks are ignored in map construction because they do not affect AUV motion. Due to the waterline of barges being shallow, the composition of the barges is poor.

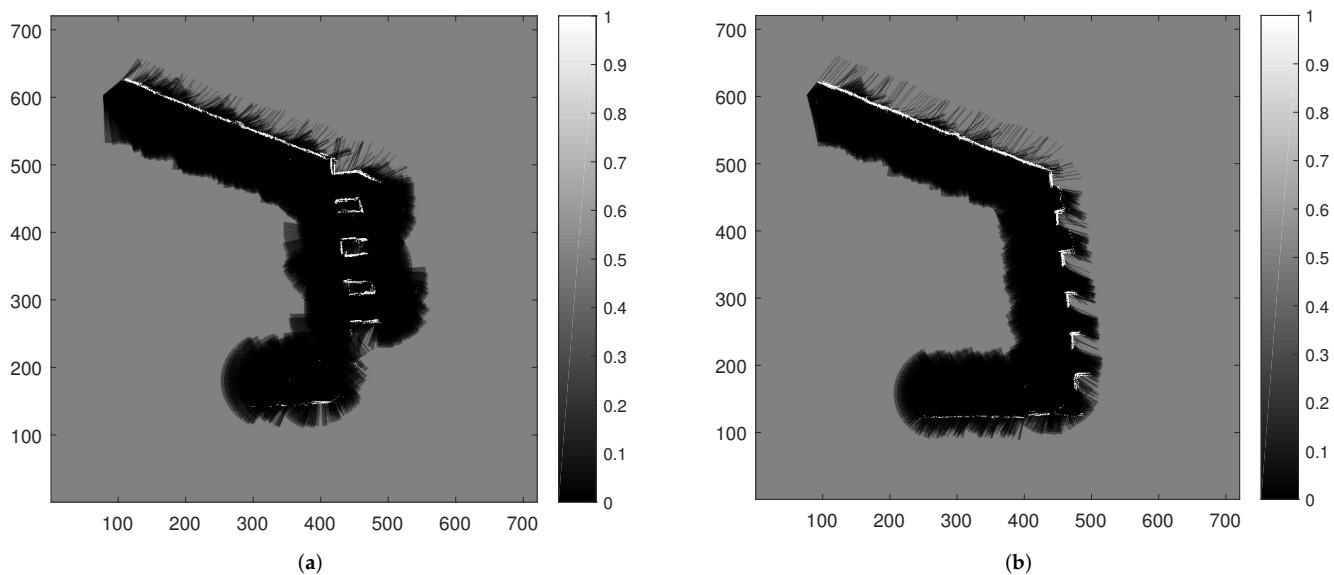

(a)                                                                                                          (b)

**Figure 12.** Probabilistic grid map of the SLAM navigation system. (**a**) Probabilistic grid map of the SLAM navigation system in Test 1. (**b**) Probabilistic grid map of the SLAM navigation system in Test 2.

*4.2. Experimental Results and Analysis under Different Filters*

The SLAM algorithm proposed in this paper is based on EKF for state estimation. As the EKF algorithm ignores the higher-order terms when calculating the Jacobian matrix, which leads to model error in the filter system, the unscented Kalman filter (UKF) was introduced to solve the problem. In this section, the UKF and the EKF SLAM algorithm are employed. The experimental results are shown in Figure 13, where black lines are the ground truth, red lines are the SLAM position estimated by EKF, and blue lines are that of UKF. According to the experimental results, the performance of SLAM under the two filters was almost the same. Although the UKF is more suitable for nonlinear systems than EKF, the nonlinearity of the proposed SLAM is weak, and the UKF has no significant improvement in this work. Considering the limitation of computational resources, the proposed EKF filter-based SLAM algorithm can meet practical applications. Figure 14 shows the localization errors of the two filters. The red lines are SLAM using EKF, and the blue lines are the errors of UKF. Table 2 shows the RMSE of AUV localization under the two filters.

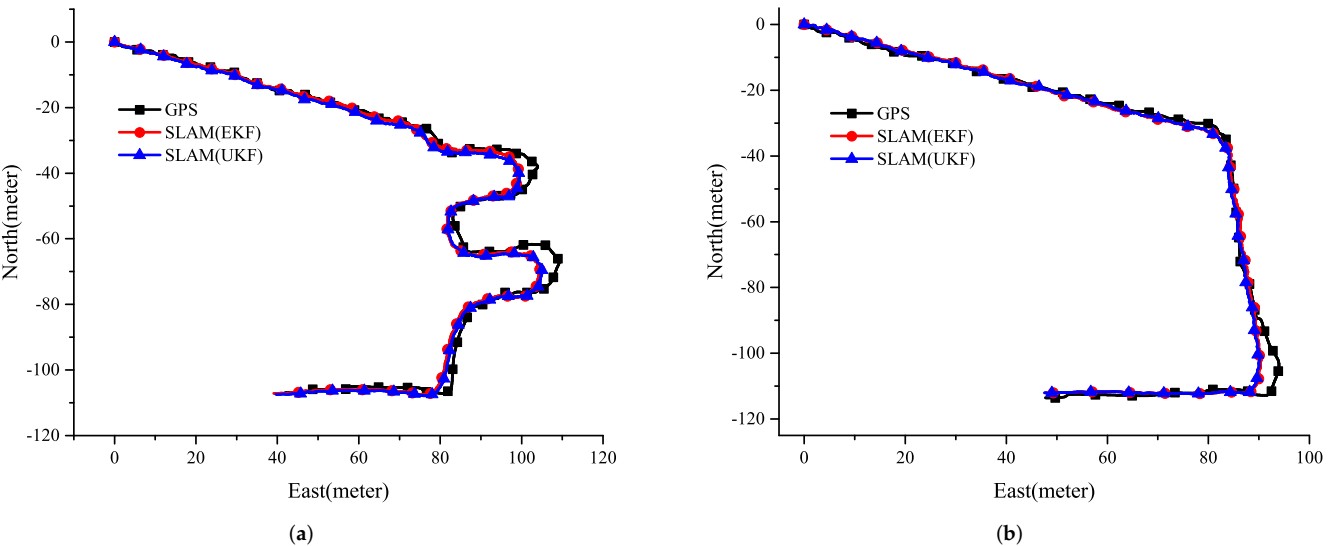

**Figure 13.** Comparison of EKF and UKF SLAM algorithms. (**a**) Comparison of EKF and UKF SLAM algorithms in Test 1. (**b**) Comparison of EKF and UKF SLAM algorithms in Test 2.

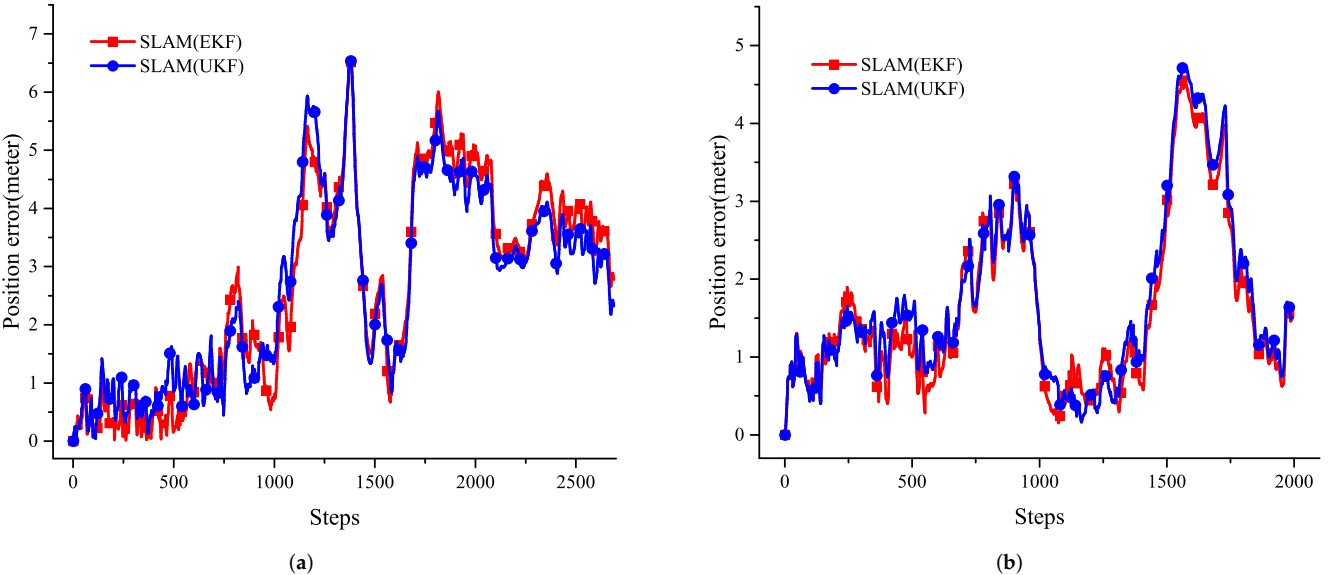

**Figure 14.** The positioning error for the EKF and UKF SLAM algorithms. (**a**) The positioning error for the EKF and UKF SLAM algorithms in Test 1. (**b**) The positioning error for the EKF and UKF SLAM algorithms in Test 2.

**Table 2.** RMSE of positioning under the EKF and UKF filters; unit: m.

|  | SLAM (EKF) | SLAM (UKF) |
| --- | --- | --- |
| Test 1 | 2.6900 | 2.6504 |
| Test 2 | 1.6108 | 1.7131 |

## 5. Conclusions

This paper proposed an occupancy grid-based SLAM algorithm for AUV. Each sonar scan is processed as a scanning grid map to match the local map extracted from the whole map. Subsequently, the PSO-GA is used to optimize the map matching process. Then, the position and orientation obtained by the matching method are used to correct the AUV



status. Lastly, the modified AUV status transforms the sonar data into the global coordinate and updates the grid map.

To verify the validity of the occupancy grid-based SLAM algorithm, we conducted a sea trial near Tuandao Bay wharf in Qingdao. The experimental results indicate that the proposed SLAM method can effectively improve the accuracy of AUV position and construct an accurate grid map. We expect that the grid map generated by our SLAM algorithm can also be a resource for path planning, obstacle avoidance and other tasks for AUV. The above algorithms were solved on a 2.9 GHz R7-4800H processor that has 16 GB ram and the GPU was NVIDIA GeForce GTX 1650. Matching a picture of sonar scan data took an average of 40 ms.

In the future, we will still focus on the AUV SLAM algorithm, and loop closure detection will be added to improve the AUV SLAM system we proposed. We will study the application of the swarm intelligence algorithm in particle filter SLAM to enhance the accuracy of underwater environment positioning and mapping further.

**Author Contributions:** Conceptualization, methodology, X.M.; software, validation, and formal analysis, X.M. and G.Y.; writing—original draft preparation, X.M.; writing—review and editing, G.Y. and N.Z.; visualization, C.C.; supervision and funding acquisition, X.M. All authors have read and agreed to the published version of the manuscript.

**Funding:** This work has been supported by a research fund from the Science and Technology on Underwater Vehicle Technology Laboratory (2021JCJQ-SYSJJ-LB06911), and the Postdoctoral Applied Research Project of Qingdao (79002002/006).

**Conflicts of Interest:** The authors declare no conflicts of interest.

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
