# Peer review of "Occupancy Grid-Based AUV SLAM Method with Forward-Looking Sonar"

_jmse, doi:10.3390/jmse10081056_

Round 1

Reviewer 1 Report

An interesting study. The authors must check the entire manuscript for adequate grammar and typos. They should seek professional help from a copyediting service. Also, the reference style of the manuscript should be 'Vancouver' which is numerical, superscript throughout the manuscript.
Please make sure the axes on the figures are described and ensure the figure captions have adequate explanations.
Make sure all tables and figures are mentioned in the text.

Author Response

Response to Reviewer 1 Comments

Point 1: An interesting study. The authors must check the entire manuscript for adequate grammar and typos. They should seek professional help from a copyediting service. Also, the reference style of the manuscript should be 'Vancouver' which is numerical, superscript throughout the manuscript.

Response 1: We thank the reviewer for his/her careful check. We have checked the full text for spelling and grammar errors and all were modified and hilighted in the attached pdf document. The reference style of the manuscript has been corrected to "Vancouver.”

Point 2: Please make sure the axes on the figures are described and ensure the figure captions have adequate explanations. Make sure all tables and figures are mentioned in the text.

Response 2: Thanks for the careful reminder of the reviewers, we have checked all the tables and figures in the manuscript and added appropriate descriptions. (Figure 5, Figure 9, Figure 10, Figure 11, Figure 12, Figure 13, Figure 14)

Reviewer 2 Report

All detailed comments are in the joint pdf file.

Tha article describes an adaptaiton of GA-PSO optimization for the grid map matching of a SLAM navigation for a planar AUV (in 2D). The litterature is well and largely mentionned, but not all the SLAM methods (as for grid map matching classical methods).  The AUV and is localization tools are correctly described, as well as the experimental setup.The use of evolutionary algorithms should be justified and compared to classical methods.

The lacks of this work are three-fold:

1) The model and method is only 2D. Even if it is most of opertaionnal situations of nowadays AUVs, the challenge today are in 3D.

2) Only kinematics is used to descibe behaviour of the AUV, while again, moste of the difficulties in AUV autonomous navigation come from dynamics (uncertainties and perturbations).

3) The main original part of this work is not enough described and studied: The evolutionary optimization. Nor GA or PSO evolution data are given, while we guess that each point of the results (Fig10-14) come from GA-PSO algorithm.

Moreover, the metho is said "pratical" but we have no clue on its integration on the onboard system (hardware/software), nor about the CPU used time.

English is globally correct but a few words are missused and some sentences are simply not well gramamtically built.

Author Response

Response to Reviewer 2 Comments

Point 1: The model and method is only 2D. Even if it is most of opertaionnal situations of nowadays AUVs, the challenge today are in 3D.

Response 1: Since the z-axis of the underwater vehicle is easily instrumented using a depth sensor, we simplify notation and consider motion only in the horizontal plane. Another reason is that the environmental perception sensor we used is two-dimensional sonar. We think reviewers' comments are precious, and in the next stage, we will expand to three-dimensional space to carry out our work.

Point 2: Only kinematics is used to descibe behaviour of the AUV, while again, moste of the difficulties in AUV autonomous navigation come from dynamics (uncertainties and perturbations).

Response 2: The AUV state is described by the kinematic, made some simplifications. But sensor data is updated over time and, to a certain extent, reflects dynamic changes(uncertainties and perturbations). We will consider the comments of the reviewers. We will add a kinetic model to describe the behavior of AUV in the next stage.

Point 3: The main original part of this work is not enough described and studied: The evolutionary optimization. Nor GA or PSO evolution data are given, while we guess that each point of the results (Fig10-14) come from GA-PSO algorithm.

Response 3: Thanks to the reviewers for their valuable suggestions. We have added relevant experimental data about the GA algorithm. But there are divergent in the PSO algorithm, and it is quite different from the real trajectory of AUV. So we did not add experimental data about the PSO algorithm.

Point 4: Moreover, the metho is said "practical" but we have no clue on its integration on the onboard system (hardware/software), nor about the CPU used time.

Response 4: Thanks to the reviewer for the reminder. We have added a description of the hardware.

To highlight the academic nature of the thesis, we decided to change the title of the paper to ‘’Occupancy grid-based AUV SLAM method with forward-looking sonar”.

Point 5: English is globally correct but a few words are missused and some sentences are simply not well gramamtically built.

Response 5: We thank the reviewer for his/her careful check. We have checked the full text for spelling and grammar errors and all were modified and marked in the attached pdf document.

For some annotations in the manuscript, the relevant responses are as follows:

Point 1: What kind of low-quality IMU do you use? Accuracy concerning accelerations and rotations is constant; only the heading measurement can drift (and not diverge) because of electromagnetic disturbances (more often because of bad calibration or localization in AUV).

Response 1: The IMU used in this manuscript is a fiber-optic gyroscope inertial measurement unit. In theory, the heading angle will not be affected by electromagnetic disturbance. Since the IMU calculates the heading angle, we think it is more accurate to use the term ‘diverge.’

Point 2: This hypothesis reduces the range of this work. The problem is much more interesting for truly 3D situations (with roll & pitch).

Response 2: This comment has been answered above.

Point 3: Parametrization of AUV state should be exeplained w.r.t. the 2D modeling (as psi is the yaw angle).

Also speeds u &v must be explicited w.r.t. the body-fixed frame AND the earth-fixed frame.

Response 2:Thanks for the careful reminder of the reviewers; the added part has been highlighted in the manuscript.

Point 3: this is dangerous to consider you control velocities and yaw rate since it depends a lot on the inertial and hydrodynamic efforts. You should consider truly controllable parameters for input control as forces and moments (unless you consider no dynamics, which again degrades the range of this work).

Response 3:It may be that the description in the manuscript is not specific, causing the reviewer to misunderstand the meaning. The control input is the control variable in the time update process in the Kalman filter, where u and v represent the forward and starboard speeds in the body-fixed frame, from DVL,  is the orientation Angle from IMU. Sensor data is updated over time and reflects dynamic changes to a certain extent. it is necessary to consider the kinetic model of AUV, and we will carry out related research in the next stage.

Point 4: something is wrong, you said the IMU diverges but you rely 100% on it for yaw angle, which is the less accurate parameter it gives.

Response 4:In fact, the yaw angle provided by the IMU does diverge, but we don't rely 100% on it. In the measurement update process, The map matching between the grid map and the local sub-map could find an appropriate AUV state, and the state is the observation in the measurement update process of the EKF. Finally, the state of the AUV that we rely on is updated through the observation vector and the system state in the time update process.

Point 5: It is not obvious this kind of problem should require GA or PSO to be solved. Authors should cite previous work on SLAM not using EA, but performing the task. What justifies such a complex solving method? Finding a matching rotation+transmation should not be so hard that a classical regression method would not work (least squares method).

Response 5: The least square method is a pure data statistics algorithm, and the matching method between the scanned map and the matched map is of non-equal magnitude. Regression methods are most effective most of the time. But the number of grids occupied by the sub-map and the constructed map may be different in some cases, so the classical regression method has a significant deviation. The branch and bound method is often adopted in the matching method, but local optimization may occur, so the algorithm in this paper is considered.

Point 6:None of the GA or the PSO algorithm is detailled, no evolution or convergence data is given so we cannot determine it worked out or not. This should be added.

Response 6:This comment has been answered above.

Point 7: It is a pity these maps orientation do not match the two in Fig5.

Response 7: Thanks to the reviewers for their careful reading. There is no connection between the two pictures, The feature extraction effect of sonar data is shown in Figure 5, and Figure 7 shows the effect of grid map matching using the proposed PSO-GA algorithm.

Point 8: if this is identity what an observer for?

Response 8: The dead reckoning algorithm is used to the state of the AUV system status. The system state is X = , and the orientation of IMU and velocity of DVL in the horizontal plane are employed, the observation vector from the map matching between the grid map and the local sub-map, the forward-looking sonar data is employed. We use the observation vector for the Kalman update considering the inaccuracy of the system state in the time update process.
